# CD4 and CD8 Lymphocyte Counts as Surrogate Early Markers for Progression in SARS-CoV-2 Pneumonia: A Prospective Study

**DOI:** 10.3390/v12111277

**Published:** 2020-11-09

**Authors:** Joan Calvet, Jordi Gratacós, María José Amengual, Maria Llop, Marta Navarro, Amàlia Moreno, Antoni Berenguer-Llergo, Alejandra Serrano, Cristóbal Orellana, Manel Cervantes

**Affiliations:** 1Rheumatology Department, Parc Taulí University Hospital, I3PT Research Institute (UAB), Universitat Autónoma de Barcelona (UAB), 08208 Sabadell, Spain; joan.calvet.fontova@gmail.com (J.C.); mariallop@hotmail.com (M.L.); corellana@tauli.cat (C.O.); 2Immunology Unit UDIAT, Parc Taulí University Hospital. I3PT Research Institute (UAB), 08208 Sabadell, Spain; mjamengual@tauli.cat; 3Infectious Disease Department, Parc Taulí University Hospital. I3PT Research Institute (UAB), 08208 Sabadell, Spain; mnavarro@tauli.cat (M.N.); mcervantes@tauli.cat (M.C.); 4Pneumology Department, Parc Taulí University Hospital, I3PT Research Institute (UAB), 08208 Sabadell, Spain; amoreno@tauli.cat; 5Biostatistics and Bioinformatics Unit, Institute for Research in Biomedicine Barcelona (IRB Barcelona), 08028 Barcelona, Spain; antonio.berenguer@irbbarcelona.org; 6Research Biology Unit, I3PT Research Institute (UAB), 08208 Sabadell, Spain; alejandra.parctauli@gmail.com

**Keywords:** COVID-19, lymphocyte subsets, CD4+ T cells, predictive value, severity

## Abstract

Background: COVID-19 pathophysiology and the predictive factors involved are not fully understood, but lymphocytes dysregulation appears to play a role. This paper aims to evaluate lymphocyte subsets in the pathophysiology of COVID-19 and as predictive factors for severe disease. Patient and methods: A prospective cohort study of patients with SARS-CoV-2 bilateral pneumonia recruited at hospital admission. Demographics, medical history, and data regarding SARS-CoV-2 infection were recorded. Patients systematically underwent complete laboratory tests, including parameters related to COVID-19 as well as lymphocyte subsets study at the time of admission. Severe disease criteria were established at admission, and patients were classified on remote follow-up according to disease evolution. Linear regression models were used to assess associations with disease evolution, and Receiver Operating Characteristic (ROC) and the corresponding Area Under the Curve (AUC) were used to evaluate predictive values. Results: Patients with critical COVID-19 showed a decrease in CD3+CD4+ T cells count compared to non-critical (278 (485 IQR) vs. 545 (322 IQR)), a decrease in median CD4+/CD8+ ratio (1.7, (1.7 IQR) vs. 3.1 (2.4 IQR)), and a decrease in median CD4+MFI (21,820 (4491 IQR) vs. 26,259 (3256 IQR)), which persisted after adjustment. CD3+CD8+ T cells count had a high correlation with time to hospital discharge (PC = −0.700 (−0.931, −0.066)). ROC curves for predictive value showed lymphocyte subsets achieving the best performances, specifically CD3+CD4+ T cells (AUC = 0.756), CD4+/CD8+ ratio (AUC = 0.767), and CD4+MFI (AUC = 0.848). Conclusions: A predictive value and treatment considerations for lymphocyte subsets are suggested, especially for CD3CD4+ T cells. Lymphocyte subsets determination at hospital admission is recommended.

## 1. Introduction

The outbreak of the novel emerging infectious disease COVID-19 associated to a newly discovered coronavirus (SARS-CoV-2) was initially detected in Wuhan, China, and it spread rapidly worldwide [1]. World Health Organization (WHO) declared a pandemic on 11 March 2020. The most prominent and leading cause of mortality and morbidity related to COVID-19 is pneumonia causing a severe acute respiratory syndrome (SARS). Although little is known about the pathophysiology of the new SARS-CoV-2, two coronaviruses causing previous outbreaks, SARS in 2002–2003 and Middle East respiratory syndrome (MERS) in 2012, might provide some insights into the COVID-19 disease [2].

The incubation period is estimated to be in a range between 4.5 and 11 days. The spectrum of symptoms differs considerably among affected people, although fever, cough, myalgia, and fatigue appear to be the most frequent. Most people suffered a mild to moderate self-limiting disease, but several cases experienced a sudden clinical deterioration 7–8 days after symptom onset, suggesting that severe respiratory failure in COVID-19 is driven by a unique pattern of immune dysfunction [3]. Up to 25% of admitted patients may require intensive care unit assistance [4].

Although current knowledge about SARS-CoV-2 pathophysiology is still limited, dysregulation of the immune system and cytokine storm have been detected, especially in critical patients. Regarding this important feature, little is known about the role of lymphocyte subsets in SARS-CoV-2 infection [5]. A study conducted in previous SARS patients found a more deeply decrease in CD3+, CD4+, and CD8+ T cell counts and the ratio CD4+/CD8+ in the early acute phase of SARS in contrast to other viruses such as HIV-1, CMV, or EBV infection, which may suggest a specific immune pathology response to coronavirus [6]. Previous works focused on lymphocyte subsets of COVID-19 found a consumption of CD4+ and CD8+ T cells, which might explain the aggravated inflammatory response, the aforementioned cytokine storm activation and worse infection prognosis [7]. Counts of lymphocyte subsets in COVID-19 patients are highly variable along the different phases of the disease and, despite the increasing literature about its associated immune response, their specific role in the SARS-CoV-2 immunopathology remains uncertain [8].

A specific treatment or a vaccine is not currently available for SARS-CoV-2 infection. Some treatments, Interleukin 6 (IL-6) inhibitor and corticosteroids, dexamethasone in particular, seem to reduce the severity of SARS-CoV-2 syndrome, especially if administered early in the course of the disease [9,10,11]. In this sense, the early identification of patients likely to benefit from their use seems to be a priority. The results of previous observational studies suggest that lymphopenia, ferritin, D-Dimer, and C-reactive protein (CRP) might be associated to a worse evolution of COVID-19 patients [12]. Unfortunately, none of them has been validated in previous series as a predictive factor for severe SARS-CoV-2 disease.

In this prospective study, we aim to evaluate the potential role of lymphocyte subsets as a candidate for severity biomarker in COVID-19 early stages and to improve therapeutic decisions.

## 2. Materials and Methods

A prospective cohort study was conducted to assess the association between COVID-19 severity and lymphocyte subsets. Study individuals were systematically selected from patients admitted to University Hospital of Parc Taulí (HUPT) between 14th April and 28th April with a confirmed COVID-19 diagnosis (positive diagnostic test based on the detection of the viral sequence by reverse transcription–polymerase chain reaction (RT-PCR) of nasopharyngeal and/or oropharyngeal swab) and confirmed typical SARS-CoV-2 lung involvement (defined as peripheral bilobar or bilateral infiltrates on chest X-ray), so all participants presented a confirmed COVID-19 pneumonia. All subjects were included before the start of any treatment specifically prescribed for COVID-19. Study inclusion was stopped at 28th April due to the favorable evolution in Spain and the dramatic drop in the number of COVID-19 patients requiring hospitalization at our center. Exclusion criteria helped to homogenize the patients sample and comprised potential confounding factors of the clinical picture, such as immunomodulatory treatments, active neoplasms in chemotherapy, age over 75 years, chronic renal failure, or patients under hemodialysis treatment, previous immunodeficiency, severe chronic obstructive pulmonary disease (FEV1 < 50%) and any opportunistic infection. All patients were suffering from a severe disease at the time of hospital admission and were remotely monitored to establish their condition during follow-up as stable or progression to critical COVID-19. Criteria for critical evolution was defined a priori as clinical features as respiratory rate ≥30 breaths per minute, PaO2 < 92% while on FiO2 ≥ 0.35, PaO2/FiO2 ratio <200, or non-invasive mechanical ventilation or orotracheal intubation requirement.

This study was approved by the Local Ethical Committee at the Hospital Universitari Parc Taulí Sabadell (2020/569, 14 April 2020). All patients were verbally informed, and witness informed consent was obtained. All methods were performed in accordance with the relevant guidelines and regulations. Authors declare that the investigations were carried out following the rules of the Declaration of Helsinki of 1975.

### 2.1. Assessments

The following information was recorded: sex, date of birth, date of symptoms onset, date of admission, date of confirmed diagnostic in emergency room (positive RT-PCR from nasopharyngeal and/or oropharyngeal swab), and cardiovascular risk factors (arterial hypertension, dyslipidemia, diabetes, and obesity) defined by established diagnosis in medical history or taking active medication. Finally, the date of worsening (when applicable) and the date of discharge from hospital were collected. Patients systematically underwent complete laboratory tests, including all parameters typically related to COVID-19 such as total leucocyte, neutrophils, lymphocytes (cells × 10^9^/L) counts, ferritin (ng/mL), CRP (mg/dL), d-Dimer (mg/mL), lactate dehydrogenase (LDH, U/L) and lymphocyte subsets study. All these measures were obtained from blood samples collected at the time of hospital admission. Information from thorax computerized tomography (CT) performed following clinical practice criteria were retrieved. Out of critical patients, six underwent a thorax CT scan. Two of them had confirmed characteristic infiltrates associated to SARS-COV-2 infection with no other complications, two demonstrated pulmonary thromboembolisms, and the other two showed definite signs of early pulmonary fibrosis.

### 2.2. Lymphocyte Subsets Determination

Lymphocyte subpopulations were analyzed in whole peripheral blood by adding a panel of monoclonal antibodies (Becton Dickinson, San Jose, CA, USA): BD Multitest™ CD3 FITC/CD16 + CD56 PE/CD45 PerCP/CD19 APC for T, B and NK subsets and BD Multitest™ CD3 FITC/CD8 APC /CD45 PerCP/CD4 APC for CD4 and CD8 T cells subsets. Sample preparation was carried out according to manufacturer guidelines. The samples were acquired by a flow cytometry analyzer (BD FACSLyric, BD Biosciences, San Jose, CA, USA) and analyzed using BD FACSuite™ Clinical Software version. Absolute numbers of cells were obtained using TruCount tubes (Becton Dickinson, San Jose, CA, USA) in the same platform. Median fluorescence intensity (MFI) of the different markers in acquired samples were analyzed with FlowJo software (FlowJo, LLC, Portland, OR, USA).

### 2.3. Statistical Methods

Summary of clinical data and laboratory parameters and their association with disease evolution were performed using non-parametric methods. Medians, interquartile ranges, and Mann–Whitney tests were used for continuous measures, while frequencies and exact Fisher’s test were applied to categorical variables. Spearman correlation coefficients (SCC) and their corresponding 95% confidence intervals (CI) were used to assess the association of blood parameters with the length of stay at the hospital.

Linear regression models were used to assess associations with outcomes when statistical control for confounders was needed (age, gender, and time from symptoms onset). Adjusted group means derived from the models and Partial Correlation Coefficients (PCC) and their corresponding 95% CI were used to display the magnitude of the effects. When needed, Tukey’s transformation was applied to the continuous variables in order to fulfill the assumptions of the linear model (see Appendix A section).

The predictive value of blood determinations and their combinations was assessed using the Receiver Operating Characteristic (ROC) and the corresponding Area Under the Curve (AUC). Predicted probabilities from a logistic regression model were used when evaluating combinations of markers. In addition, blood determinations were simultaneously evaluated and prioritized according to their predictive power of disease evolution in an agnostic manner. For doing so, we used logistic regression via LASSO penalization of the maximum likelihood (M1), as implemented in the R package glmnet [M2] (see Appendix A section). LDH was excluded from the later analysis because values were not available for one-third of the patients [10] in this determination. In order to avoid model overfitting, markers combinations were evaluated using a leave-one-out cross-validation process. Intervals at 95% confidence were computed for AUCs using bootstrap [M3].

A 5% was set as the threshold for statistical significance. All statistical analyses were conducted with R [M4] (see Appendix A section for extended details of the Statistical Methods)

## 3. Results

A total of thirty patients were recruited following the inclusion criteria. During follow-up, thirteen patients (43.3%) progressed to critical disease condition according to the criteria previously established. Individual characteristics at admission time are described in Table 1. Selected patients were mostly male (67%), median age was 60.6 (7.2 IQR), and median time from symptoms onset to hospital admission was 7 days (4 IQR). Prevalence of cardiovascular risk factors was low in our series (20% for hypertension, 17% for dyslipidemia, and 3% for diabetes and obesity). No differences between the two groups were found for any of these mentioned features (Table 1). A 34% increase in total lymphocytes count and a 40% decrease in levels of d-Dimer in men compared to women were the only parameters at admission showing statistical differences regarding gender (Appendix A). It should be noted that 12 of 13 critical patients were treated with tocilizimab, and all 13 received corticosteroids. Only one patient of the critical group died. All of the patients in the non-critical group were discharged without clinical complications in a median of 5 days from hospitalization, and none of them were treated with corticosteroids nor tocilizumab. All the therapeutic decisions were performed for the clinicians responsible of the patient without any interference of the study group.

Concentrations of lymphocyte subsets were quantified at the admission time to the hospital in order to identify differences between disease evolution (Table 1). Patients who reached a critical disease condition showed decreased CD3+CD4+ T cells count, CD4+/CD8+ ratio, and CD4+MFI compared to individuals with persistent condition. Specifically, critical patients compared to non-critical showed a 49% decrease in the median CD3+CD4+ T cells count (278 (485 IQR) vs. 545 (322 IQR)), a 45% decrease in median CD4+/CD8+ ratio (1.7, (1.7 IQR) vs. 3.1 (2.4 IQR)), and a 17% decrease in the median of CD4+MFI (21,820 (4491 IQR) vs. 26,259 (3256 IQR)). These associations reached statistical significance even after control by potential confounders such as age, sex, and time from symptoms onset to hospital admission (Table 1 and Table 2 and Figure 1). In addition, after control for confounders, increased levels of ferritin (2-fold change) and LDH (27%) were observed in the critical group compared to the rest of patients, although none of these comparisons reached statistical significance (Table 2). No association with disease evolution was found for any of the other blood parameters typically used in clinical practice for COVID-19 (leucocytes, neutrophils, lymphocytes, C-reactive protein, and d-Dimer) or any of the other lymphocyte subset quantification (Table 1 and Table 2). Interestingly, the two patients with pulmonary fibrosis showed the lowest CD3+CD4+ and CD3+CD8+ T cells count at admission time (Appendix A).

To gain further insight into factors influencing COVID-19 prognosis, we used length of hospital stay as a surrogate of severity in disease evolution. Interestingly, estimated correlations for several blood parameters were highly dependent on the severity condition group (Appendix A). In patients who progressed to critical condition, CD3+CD8+ T cells count was the only parameter (negatively) correlated with time of hospital discharge (PCC = −0.700 (−0.931, −0.066)) (Figure 2). In patients with a persistent disease, CD4+MFI showed a strong negative correlation with time until recovery after control for confounders (PCC = −0869 (−0.958, −0.627)) (Figure 2). In addition, in non-critical patients, time to discharge was positively correlated with age (PCC = 0.548 (0.049, 0.828)) and ferritin (PCC= 0.463 (−0.089, 0.798), and it was negatively correlated with lymphocyte count (PCC = −0.459 (−0.796, 0.095)), although the latter two did not reach statistical significance (Appendix A).

Finally, to assess their value for the prediction of disease evolution, ROC curves were independently computed for each single determination, which was then evaluated according to their AUC values. Blood parameters usually chosen for COVID-19 severity monitoring (lymphocytes, ferritin, CRP, LDH, and d-Dimer) showed AUC values ranging between 0.468 and 0.682 (Appendix A). Regarding measurements of lymphocyte subsets, best performances were achieved by CD3+CD4+T cells (AUC = 0.756), CD4+/CD8+ ratio (AUC = 0.767), and CD4+MFI (AUC = 0.848) (Figure 3). A logistic regression model combining these three aforementioned parameters did not show any substantial improvement of prediction performance (AUC = 0.837) (Appendix A). When all blood measurements were simultaneously evaluated in a multivariate setting, CD3+CD4+ T cells, CD4+/CD8+ ratio, and CD4+MFI were the only markers selected by the models in all 30 instances of a leave-one-out cross-validation procedure, which highlighted the predictive power of these parameters and indicated that, to some extent, they contribute independently to the prediction of COVID-19 critical evolution. Probabilities derived from the cross-validation instances provided a 0.765 AUC (Appendix A, Appendix A).

Cell counts were used for all the previous analyses involving lymphocytes subsets evaluations. Association analyses for lymphocyte subsets percentages yielded similar results and are provided in Appendix A.

## 4. Discussion

To our knowledge, this is the first prospective study in Caucasian patients, specifically focused in an early phase of SARS-CoV-2 severe infection. This study, which included patients at time of admission, showed a significantly decrease in the total CD3+CD4+ T cells count, in the CD4+/CD8+ ratio, and in the CD4+MFI, which were linked to a critical evolution of COVID-19 patients with bilateral pneumonia. In this sense, all of the results reported suggest that CD3+CD4+ T cells count and CD4+ levels of expression may be essential for the early detection of SARS-CoV-2 critical evolution and to select patients candidates to promptly receive aggressive treatments with corticosteroids or IL-6 inhibitors. In our study all critical patients received these treatments during the follow up while they were not administered to any of the patients in the non-critical group, so a therapeutic consideration might be predicted by lymphocyte subsets.

Immune system dysregulation, comprising lymphopenia and a cytokine storm, has been previously observed in SARS-CoV and appears to be associated with the severity of the infection [13]. There are two key features that may describe the immune dysregulation associated to COVID-19. One is an overproduction of pro-inflammatory cytokines by monocytes, and the other is a dysregulation of lymphocytes characterized by CD4 lymphopenia and subsequently B cell lymphopenia [3]. The cellular immune response activation has been shown to be the most effective against viral infections [14]. Severe SARS-CoV infection in humans is characterized by the delayed development of the adaptive immune response and delayed virus clearance [15].

Previous retrospective studies focused on T lymphocyte subsets in COVID-19 disease found a decrease in CD3+, CD4+, and CD8+ T cells median count in critical patients [16]. Moreover, total lymphocytes, CD4+, and CD8+ T cells decreased in COVID-19 patients compared to healthy controls, with severe cases showing lower levels compared to the rest [17]. In contrast to our work, most of these studies were performed retrospectively in late phases of the disease. In this sense, these previous studies do not seem to present an adequate approach to evaluate the specific role of the different lymphocyte subsets in early stages of the disease as a prognostic factor of severity.

Our results are in accordance with the previously published data in SARS-CoV infection in animal models. In those studies, the immune T cell response was responsible for virus clearance [18]. Specifically, the CD4 T cell depletion resulted in a decrease of neutralizing antibodies and lung virus clearance. Conversely, CD8 T cell depletion in early phases did not affect neutralizing antibodies or virus clearance, suggesting that a CD4-dependent virus-specific response is critical to control SARS-CoV infection [19].

In our study, the number of hospitalization days was used as a surrogate of disease severity. Interestingly, the data we observed pointed to an inverse association between CD3+CD8+ T cells and CD4+MFI and the number of days of hospitalization. The results reported were not unexpected, since we evidenced that the number CD3+CD4+ T cells and the membrane expression levels of CD4+ were associated to critical disease. Regarding CD3+CD8+ T cells, retrospective studies performed in patients with severe disease reported a deep decrease in CD8+ T cells [20]. Additionally, a significant association between low CD8 T cells count and ICU requirements was described [21]. In accordance with these data, we observed that the CD3+CD8+ T cell levels were linked to longer time to discharge, especially in critical patients. An important decrease not only in CD4+, but also in CD8+ T lymphocyte count was previously reported in critical patients, pointing to the importance of a dysregulated immune response in COVID-19 pathogenesis [22]. In a similar way to chronic infections, COVID-19 might damage CD4+ T cells function and promote the excessive activation of CD8+ T cells with the subsequent potential exhaustion [23]. The results of a recent prospective study in hospitalized Covid-19 patients are in accordance with our data with CD4+ and CD8+ T cells decreased with a later reconstitution of CD8+ T cells [24].

The downregulation of CD4 membrane expression, recorded by CD4+MFI, has long been reported in other viral diseases such as HIV or herpesvirus infection and, in some instances, it could be related to the endocytosis of the viral particle that involves the aggregation and internalization of lipid rafts. This hypothesis is supported by a recent study showing that SARS-CoV-2 could infect T cells through receptor-dependent or S protein-mediated membrane fusion [25].

Unlike most previous reports, we observed a decrease in CD3+CD4+ T cells, CD4+/CD8+ ratio, and CD4+MFI with normal levels and the expression of CD3+CD8+ T cells. These differences might be explained by the characteristics of our study, which are prospective and focused on the early disease phase in severe patients who needed hospitalization (samples were obtained upon admission to hospital). Overall, the results of our study may contribute to the understanding of the physiopathology of SARS-CoV-2 infection, suggesting an initial involvement of CD3+CD4+ T cells and, probably, a later participation of CD3+CD8+ T cells, suggesting a more precise role of CD3+CD8+ T cells in terms of disease chronicity. In accordance with this hypothesis, a previous study in intensive care unit patients reported an increase in CD8+MFI, which could indicate a higher cytotoxic activity through the overexpression of CD8 protein [26]. In addition, although peripheral CD8 T cells were reduced in COVID-19, ARDS, an over-activation in the respiratory tract manifested by higher cytotoxicity activity and inflammatory response, might partially explain the severe immune injury observed in those patients [27].

To find predictive factors for COVID-19 severity is a priority during this severe outbreak. Several previous retrospective studies have addressed this subject and reported different models supported by clinical parameters or by a combination of clinical and laboratory factors [28]. Among them, age, neutrophil to lymphocyte ratio, serum levels of CRP, IL6, D-Dimer, and comorbidities are the most recommended [29,30,31]. A risk stratification test was even developed in this context to classify patients as low, intermediate, or high risk for ICU requirements. The test is composed by clinical characteristics such as respiratory rate and medical history of cardiovascular heart disease, and laboratory parameters such as PaO_2_ to FiO_2_ ratio, CRP, and creatinine levels [12]. Moreover, in one previous study, CD3+CD8+ T cells, together with cardiac troponin I, were used as mortality predictive factors in COVID-19 pneumonia patients [32]. However, none of these have showed a great accuracy to predict severe COVID-19, none of them has been evaluated specifically in a prospective cohort study, and no treatment implications were evaluated. Given the characteristics of our study and the data observed for CD3+CD4+ T cells at hospital admission, we decided to assess the performance of lymphocyte subsets as a predictive factor for critical COVID-19. The CD3+CD4+ T cells count and the CD4+MFI showed an accuracy of 0.756 and 0.848 respectively as predictive factors measured by AUC, which did not improve when combining with other factors. Interestingly, when all blood measurements were simultaneously evaluated in a multivariate setting, CD3+CD4+ T cells, CD4+/CD8+ ratio, and CD4+MFI were the only markers selected by the models in all 30 instances of a leave-one-out cross-validation procedure, highlighting the predictive power of these parameters and indicating that to some extent, they might contribute independently to the prediction of COVID-19 evolution. These observed results appear to be definitively better than the classic parameters previously proposed as predictive factors. In this sense, currently available therapy for SARS-CoV-2 is usually started late in the course of the disease. In addition, all of our patients in the critical group received corticosteroids or tocilizumab treatment, as they would be considered as a selected group for a prompt start of aggressive treatments. Thus, our findings could contribute to improve therapeutic decisions and therefore might prevent evolution to critical disease.

The sample size of our study makes the results provisional, and further larger confirmatory studies are required. However, all the results obtained in the different subanalyses performed are consistent, showing an association between CD4+ T cells and disease severity in the early stages of COVID-19. The exclusion of people older than 75 could be a limitation that prevents extrapolating conclusions to the standard population. On the other hand, it is to be noted that this study was performed on a very homogeneous sample of SARS-CoV-2 patients, which allowed us to observe differences of a mild to moderate magnitude even with a modest sample size.

In conclusion, the data we observed support a specific role of CD4 T lymphocytes in early stages and a potential implication of CD8 T lymphocytes in later stages of COVID-19. These results suggest the relevance of including lymphocyte subsets determinations at admission in order to improve the management and therapeutics of these patients.

## Figures and Tables

**Figure 1 viruses-12-01277-f001:**
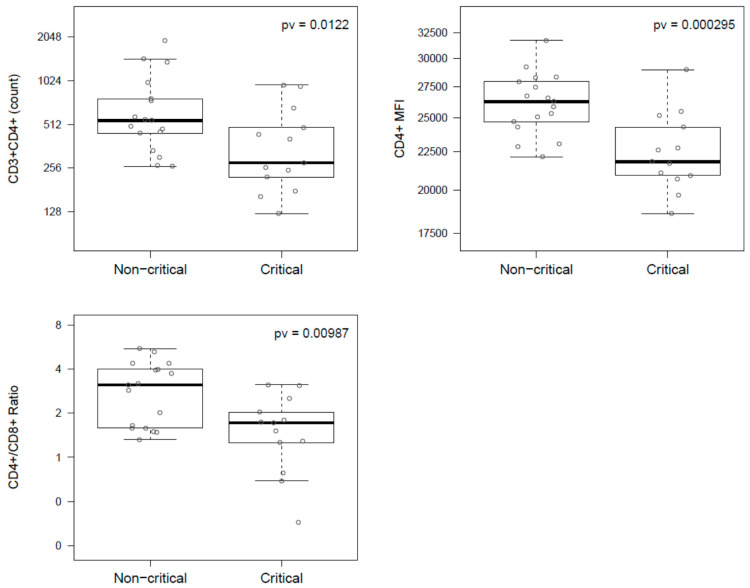
Boxplot showing association of CD3+CD4+ count (top-left), CD4+MFI (top-right), and CD4+/CD8+ ratio (bottom-left) with COVID-19 evolution. *p*-values were derived from the F-test of a linear model.

**Figure 2 viruses-12-01277-f002:**
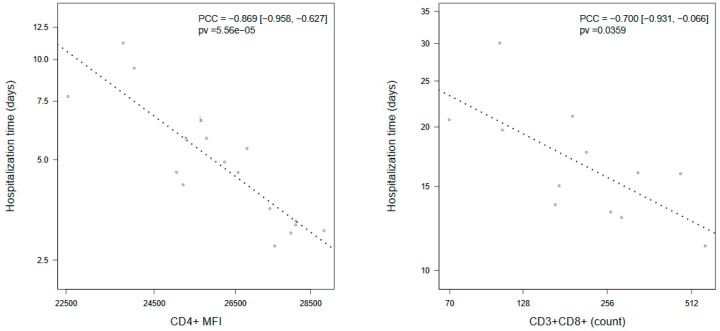
Dotplots showing association of stay length in hospital with: CD4+MFI in COVID-19 patients that did not change non-critical clinical status (CD4 + MFI); and CD3+CD8+ T cell count in patients who reached critical condition during their hospitalization (CD3+CD8+ count). Association was assessed using a linear model controlling by age, gender, and time from symptoms onset. *p*-values were derived from a F-test of the linear model. Association was measured using Partial Correlation Coefficients (PCC) and their 95% confidence intervals. Dot lines represent the slope estimated by the linear model. Values are displayed after a-priori correction by the rest of covariates in the model.

**Figure 3 viruses-12-01277-f003:**
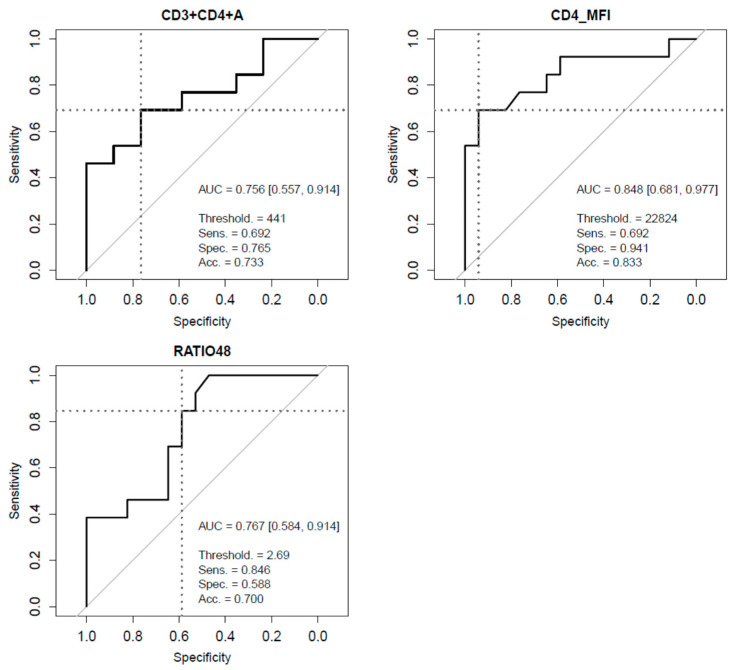
Receiver Operating Characteristics (ROC) and their corresponding Area Under the Curve (AUC) assessing the ability of CD3+CD4+T cell count (CD3+CD4+A), CD4+MFI (CD4_MFI) and CD4+/CD8+ ratio (RATIO48) to predict COVID-19 critical clinical evolution. AUC intervals at 95% confidence were computed using bootstrap. Total accuracy, sensitivity, and specificity are displayed for the optimal threshold, defined as the ROC point closest to the top-left part of the plot (perfect sensitivity and specificity).

**Table 1 viruses-12-01277-t001:** Patients characteristics and blood measurements at time of admission and their univariate association with COVID-19 evolution.

Patients Characteristics and Blood Measurements	All (*n* = 30)	Non-Critical (*n* = 17)	Critical (*n* = 13)	*p*-Value
Age	60.6 (6.1, 63.3)	60.1 (51.7, 74.9)	61.1 (55.2, 64.5)	0.9833
Gender (Male)	20 (66.7%)	12 (70.6%)	8 (61.5%)	0.6030
Days of symptoms onset	7.000 (6.000, 10.000)	7.000 (4.000, 11.000)	6.000 (5.000, 10.000)	0.6439
Days to hospital discharge	8.000 (5.000, 14.000)	5.000 (4.000, 6.000)	15.500 (12.000, 22.000)	<0.001
HT	6 (20.0%)	3 (17.6%)	3 (23.1%)	0.7134
DM	1 (3.3%)	1 (5.9%)	0 (0.0%)	0.2810
DLP	5 (16.7%)	2 (11.8%)	3 (23.1%)	0.4119
OBESITY	1 (3.3%)	0 (0.0%)	1 (7.7%)	0.1900
Leucocyte count (cells × 10^9^/L)	6310 (5310, 8860)	6550 (5310, 9440)	5970 (5120, 11370)	0.4512
Neutrophyl count (cells × 10^9^/L)	4440 (3920, 6650)	4570 (3950, 7030)	4200 (2900, 9370)	0.4388
Lymphocyte count (cells × 10^9^/L)	1215 (1040, 1310)	1260 (1040, 1440)	1180 (920, 1840)	0.5030
Ratio N/L	4.26 (3.05, 5.08)	4.19 (2.90, 5.08)	4.33 (1.58, 7.95)	0.8835
Ferritin (ng/mL)	711.7 (382.6, 1136.2)	639.7 (270.6, 1136.2)	783.7 (354.5, 2390)	0.2330
CRP (mg/dL)	8.80 (5.07, 11.25)	8.54 (4.74, 11.25)	9.50 (5.00, 15.64)	0.3909
D-Dimer (mg/mL)	691 (443, 860)	703 (443, 860)	679 (269, 1722)	0.7695
LDH (U/L)	282 (244, 365)	267 (238, 387)	356 (243, 446)	0.1713
T lymphocyte count	714 (497, 823)	725 (497, 1119)	647 (375, 1113)	0.4025
CD3+CD4+ count	467 (303, 574)	545 (445, 767)	278 (178, 663)	0.0180
CD3+CD8+ count	245 (171, 319)	253 (145, 319)	237 (87, 586)	0.7064
CD3+CD4+CD8+ count	13 (8, 21)	16 (9, 24)	11 (4, 35)	0.295
CD3+CD4−CD8− count	18.000 (12.000, 23.000)	19 (12, 27)	12 (5, 23)	0.2249
B Lymphocyte count	112 (78, 162)	121 (86, 185)	79 (46, 197)	0.3254
Natural Killer count	196 (154, 253)	192 (140, 278)	234 (128, 327)	0.8017
Ratio CD4+/CD8+	1.91 (1.58, 3.12)	3.12 (1.58, 3.99)	1.72 (0.78, 2.52)	0.0135
CD4+ MFI	24861 (22770, 26259)	26259 (24683, 27939)	21820 (20666, 25157)	0.0013
CD8+ MFI	25856 (23819, 27476)	25948 (23819, 27607)	25337 (22878, 32176)	0.7855

HT: Arterial Hypertension; DM: diabetes; DLP: dyslipidemia; Ratio N/L: ratio Neutrophil to Lymphocyte; CRP: C-reactive protein; LDH: lactate dehydrogenase; MFI: median fluorescence intensity. Group medians and percentiles 25 and 75 (continuous variables) or absolute and relative frequencies (categorical variables) are showed. *p*-values are derived from a Mann–Whitney test (continuous variables) or an exact Fisher’s test (binary variables).

**Table 2 viruses-12-01277-t002:** Association of blood determinations with COVID-19 evolution after statistical control by age, gender, and time from symptoms onset.

Blood Determinations	Adjusted Means (95% CI)		F-Test
	Non-Critical	Critical	*p*-Value
Leucocyte count (cells × 10^9^/L)	7292.5 (5851.2, 9088.9)	6789.8 (5275.6, 8738.5)	0.6665
Neutrophyl count (cells × 10^9^/L)	5167.1 (3921.5, 6808.3)	4871.5 (3551.3, 6682.3)	0.7764
Lymphocyte count (cells × 10^9^/L)	1281.7 (1081.5, 1519.0)	1209.2 (995.3, 1468.9)	0.6487
Ratio N/L	4.03 (2.89, 5.62)	4.03 (2.75, 5.91)	0.9980
Ferritin (ng/mL)	485.2 (290.9, 809.1)	981.9 (546.5, 1764.2)	0.0757
CRP (mg/dL)	7.37 (4.56, 10.84)	8.93 (5.41, 13.33)	0.5285
D-Dimer (mg/mL)	573.9 (426.2, 814.4)	588.6 (418.3, 888.8)	0.9175
LDH (U/L)	268.5 (229.0, 319.2)	341.4 (280.2, 425.1)	0.0776
T lymphocyte count	829.3 (606.8, 1086.5)	683.3 (456.6, 955.6)	0.3991
CD3+CD4+ count	597.8 (445.8, 801.6)	331.5 (236.9, 464.0)	0.0122
CD3+CD8+ count	214.8 (153.1, 301.5)	217.2 (147.3, 320.3)	0.9659
CD3+CD4+CD8+ count	15.4 (9.8, 24.3)	11.5 (6.8, 19.5)	0.4027
CD3+CD4−CD8− count	19.3 (12.4, 30.1)	10.7 (6.4, 17.7)	0.0840
B Lymphocyte count	129.7 (92.5, 173.3)	101.9 (65.0, 147.0)	0.3356
Natural Killer count	198.0 (148.9, 254.1)	199.4 (143.5, 264.4)	0.9725
Ratio CD4+/CD8+	2.65 (2.01, 3.50)	1.49 (1.08, 2.05)	0.0010
CD4+ MFI	26128 (24878, 27441)	22416 (21192, 23712)	0.0003
CD8+ MFI	26076 (23953, 28386)	25863 (23465, 28506)	0.8980

Ratio N/L: ratio neutrophil to lymphocyte; CRP: C-reactive protein, LDH: lactate dehydrogenase; MFI: median fluorescence intensity; PCC: Partial Correlation Coefficient; CI: confidence interval. Adjusted group means, their 95% confidence intervals (CI) and F-test *p*-values derived from a linear model are showed.

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
