# Peer review of "CD4 and CD8 Lymphocyte Counts as Surrogate Early Markers for Progression in SARS-CoV-2 Pneumonia: A Prospective Study"

_viruses, 2020, doi:10.3390/v12111277_

Round 1
Reviewer 1 Report
This is a well designed and well written study aimed to identify biomarkers specific for illness severity in the early phase of the disease. The design and the applied statistic analyses are of very good quality.
Although the number of patients is low, these results may suggest further studies.
I really recommend the publication.
Author Response
Thank you for your comments.
Reviewer 2 Report
In this article, Calvet et al. characterized PBMCs of confirmed positive COVID-19 patients admitted to their medical center. COVID-19 disease progression was monitored by blood measurements relative to those taken at the time of admission for patients in this study. Using criteria established prior to the start of this study, patients were binned into two groups, non-critical or critical, based on their COVID-19 disease progression. No statistical differences in blood chemistries were observed between non-critical and critical patients. However, significant differences were observed for CD3+CD4+ T cell counts, CD4+/CD8+ T cell ratio, and CD4+ T cell MFI between the two groups. While this article is informative and overall sound, several minor points should be addressed below:
- Line 256-257 – the authors offer one potential explanation for progression to severe disease being CD4+ T cell lymphopenia and subsequently B cell lymphopenia. While this is logical, no difference in B cell numbers were observed between non-critical and critical groups within this study. The authors should clarify this statement.
- The authors should state of any potential effects (or lack thereof) tocilizimab or corticosterioid treatments may have had on CD3+CD4+ T cell counts, CD4+/CD8+ T cell ratio, or CD4+ T cell MFI on the critical cohort (i.e. could these treatments have affected the overserved results in any way).
- The authors should speculate on the potential role of PD1-PDL in the observed CD4+ T cell lymphopenia in critical patients.
- In lines 62-65, the authors link decreased T cell counts to infection with coronaviruses in previous studies. In this study, the median age in this study was 60.6 years. Could the authors state the observed effect of infection with other respiratory viruses such as influenza virus, respiratory syncytial virus, or parainfluenza virus on individuals in this age range?
- Using the data at hand, can the authors conclude a specific threshold value for CD3+CD4+ T cell counts, CD4+/CD8+ T cell ratio, or CD4+ T cell MFI that is indicative severe disease? This would be beneficial to clinicians treating COVID-19 patients.
Author Response
In this article, Calvet et al. characterized PBMCs of confirmed positive COVID-19 patients admitted to their medical center. COVID-19 disease progression was monitored by blood measurements relative to those taken at the time of admission for patients in this study. Using criteria established prior to the start of this study, patients were binned into two groups, non-critical or critical, based on their COVID-19 disease progression. No statistical differences in blood chemistries were observed between non-critical and critical patients. However, significant differences were observed for CD3+CD4+ T cell counts, CD4+/CD8+ T cell ratio, and CD4+ T cell MFI between the two groups. While this article is informative and overall sound, several minor points should be addressed below:
- Line 256-257 – the authors offer one potential explanation for progression to severe disease being CD4+ T cell lymphopenia and subsequently B cell lymphopenia. While this is logical, no difference in B cell numbers were observed between non-critical and critical groups within this study. The authors should clarify this statement.
Thank you for this interesting observation. In this study, all samples were collected and analysed in an early phase of the infection, at the time of admission to the hospital. Patients with low levels of B lymphocytes reported in previous studies were in a critical situation presenting with severe respiratory failure with distress respiratory syndrome (Complex Immune Dysregulation in COVID-19 Patients with Severe Respiratory Failure; Evangelos J. Giamarellos-Bourboulis et al; Cell Host Microbe. 2020 Jun 10; 27(6): 992–1000 and Pro- and Anti-Inflammatory Responses in Severe COVID-19-Induced Acute Respiratory Distress Syndrome—An Observational Pilot Study; Quirin Notz et al, Front. Immunol., 06 October 2020). As patients in our study were in an earlier phase of the infection, it would be possible that B lymphocytes were associated in a more delayed phase of infection.
- The authors should state of any potential effects (or lack thereof) tocilizimab or corticosterioid treatments may have had on CD3+CD4+ T cell counts, CD4+/CD8+ T cell ratio, or CD4+ T cell MFI on the critical cohort (i.e. could these treatments have affected the overserved results in any way).
Patients included in this study did not receive any medication before blood extraction. In order to clarify this, a sentence was added in the manuscript, line 89.
- The authors should speculate on the potential role of PD1-PDL in the observed CD4+ T cell lymphopenia in critical patients.
We really appreciate the reviewer’s comment. PD1-PDL is involved in different kind of tumours and nowadays in some immunomediated therapies. There is a reference of a previous study, where PD1-PDL blockade was used as a therapeutic option for visceral leishmaniasis. In this sense, the exhaustion of CD4 is associated to a worse visceral leishmaniasis and it seems that with this treatment, an improvement was observed preventing T cell exhaustion (PDL-1 Blockade Prevents T Cell Exhaustion, Inhibits Autophagy, and Promotes Clearance of Leishmania donovani, Samar Habib, Infect Immun . 2018 May 22;86(6):e00019-18). Although we consider this option as a possible explanation for cell exhaustion in COVID-19 infected patients, our study was focused in biomarkers for early detection of candidates for severe disease and in our opinion, this speculative way of action implies a more detailed explanation in the manuscript. We will consider this option in future studies.
- In lines 62-65, the authors link decreased T cell counts to infection with coronaviruses in previous studies. In this study, the median age in this study was 60.6 years. Could the authors state the observed effect of infection with other respiratory viruses such as influenza virus, respiratory syncytial virus, or parainfluenza virus on individuals in this age range?
We thank the reviewer for this interesting question. As you could observe in reference 6, the decrease of CD4 experienced by patients suffering from SARS-CoV infection was greater than HIV in early stage, EBV and CMV. Patient age in that study was 40 years old, and the profile of CD4 was similar to that in our study, confirming the hypothesis of a decrease of CD4 in SARS-CoV infection. Other viruses with similar conduct are measles and respiratory syncytial virus in children. We could not provide more detailed information regarding age.
- Using the data at hand, can the authors conclude a specific threshold value for CD3+CD4+ T cell counts, CD4+/CD8+ T cell ratio, or CD4+ T cell MFI that is indicative severe disease? This would be beneficial to clinicians treating COVID-19 patients.
We agree with the reviewer that a threshold value for lymphocyte subsets might be informative and of interest for clinicians treating COVID-19 patients. This information (for our sample of patients) is stated in figure 3. Because of the nature of this study, with a small number of patients, we prefer to emphasize the direction of the associations and not to give a selective cut of values, and leave this cut of values for validation studies. We have decided not to include this information in results section or discussion in order to be cautious with the results, and not to induce changes in clinical practice without larger and confirmatory studies. Anyway, if reviewer or editor consider it of interest, we could include it in the manuscript.
